# Verification of Generative-Model-Based Visual Transformations

## Abstract

Generative networks are promising models for specifying visual transformations. Unfortunately, certification of generative models is challenging as one needs to capture sufficient non-convexity so to produce precise bounds on the output. Existing verification methods either fail to scale to generative networks or do not capture enough non-convexity. In this work, we present a new verifier, called APPROXLINE, that can certify non-trivial properties of generative networks. APPROXLINE performs both deterministic and probabilistic abstract interpretation and captures infinite sets of outputs of generative networks. We show that APPROXLINE can verify interesting interpolations in the network's latent space.

## 1 Introduction

Neural networks are becoming increasingly used across a wide range of applications, including facial recognition and autonomous driving. So far, certification of their behavior has remained predominantly focused on uniform classification of norm-bounded balls (Gehr et al., 2018; Katz et al., 2017; Wong et al., 2018; Gowal et al., 2018; Singh et al., 2018; Raghunathan et al., 2018; Tjeng et al., 2017; Dvijotham et al., 2018b; Salman et al., 2019; Dvijotham et al., 2018c; Wang et al., 2018), which aim to capture invisible perturbations.

However, a system's safety can also depend on its behavior on visible transformations. For these reasons, investigation of techniques to certify more complex specifications has started to take place (Liu et al., 2019; Dvijotham et al., 2018a; Singh et al., 2019). Of particular interest is the work of Sotoudeh & Thakur (2019) which shows that if the inputs of a network are restricted to a line segment, the verification problem can sometimes be efficiently solved exactly. The resulting method has been used to certify non-norm-bounded properties of ACAS Xu networks (Julian et al., 2018) and improve Integrated Gradients (Sundararajan et al., 2017).

**This work** We extend this technique in two key ways: (i) we demonstrate how to soundly approximate EXACTLINE, handling significantly larger networks faster than even methods based on sampling can (a form of deterministic abstract interpretation), and (ii) we use this approximation to provide *guaranteed bounds* on the probabilities of outputs given a distribution over the inputs (a form of probabilistic abstract interpretation). We believe this is the first time probabilistic abstract interpretation has been applied in the context of neural networks. Based on these techniques, we also provide the first system capable of certifying interesting properties of generative networks.

**Main contributions** Our key contributions are:

- A verification system APPROXLINE, capable of flexibly capturing the needed non-convexity.
- A method to compute tight deterministic bounds on probabilities with APPROXLINE, which is to our knowledge the first time that probabilistic abstract interpretation has been applied to neural networks.
- An evaluation on autoencoders for CelebA, where we prove for the first time the consistency of image attributes through interpolations.
- The first demonstration of deterministic verification of certain visible, highly non-convex specifications, such as that a classifier for "is bald" is robust to different amounts of "moustache," or that a classifier $n_A$ is robust to different head rotations, as shown in Figure 1.

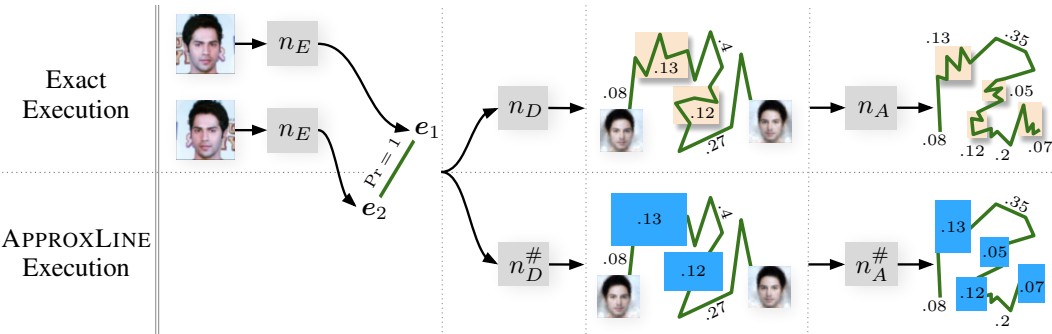

Figure 1: Using APPROXLINE to find probability bounds for a generative specification over flipped images. Green polygonal chains represent activation distributions at each layer exactly. Blue boxes are relaxations of segments highlighted by yellow boxes. We label regions with their probabilities.

## 2 OVERVIEW

Here, we introduce the terminology of robustness verification and provide an overview of our verification technique. Let $N \colon \mathbb{R}^m \to \mathbb{R}^n$ be a neural network with $m$ inputs and $n$ output classes which classifies an input $\boldsymbol{x} \in \mathbb{R}^m$ to class $\arg\max_i N(\boldsymbol{x})_i$.

**Specification** A robustness specification is a pair $(\mathbb{X}, \mathbb{Y})$ where $\mathbb{X} \subseteq \mathbb{R}^m$ is a set of input activations and $\mathbb{Y} \subseteq \mathbb{R}^n$ is a set of permissible outputs for those inputs.

**Deterministic robustness** Given a specification $(\mathbb{X}, \mathbb{Y})$, a neural network $N$ is said to be $(\mathbb{X}, \mathbb{Y})$-robust if for all $\boldsymbol{x} \in \mathbb{X}$, we have $N(\boldsymbol{x}) \in \mathbb{Y}$. In the adversarial robustness literature, the set $\mathbb{X}$ is usually an $l_2$- or $l_\infty$-ball, and $\mathbb{Y}$ is a set of outputs that correspond to a specific classification. In our case, $\mathbb{X}$ shall be a line segment connecting two encodings. The *deterministic verification problem* is to prove (ideally with $100\%$ confidence) that a network is deterministically robust for a single specification. As deciding robustness is NP-hard (Katz et al., 2017), the problem is frequently relaxed to permit false negatives (but not false positives) and solved by sound *overapproximation*.

**Probabilistic robustness** Even if $N$ is not completely robust, it may still be useful to quantify its lack of robustness. Given a distribution $\mu$ over $\mathbb{X}$, we are interested in finding provable bounds on the *robustness probability* $\Pr_{\boldsymbol{x} \sim \mu}[N(\boldsymbol{x}) \in \mathbb{Y}]$, which we call *probabilistic [robustness] bounds*.

### 2.1 VERIFYING INTERPOLATION SPECIFICATIONS

It is well known that certain generative models appear to produce interpretable transformations between outputs for interpolations of encodings in the latent space (Dumoulin et al., 2016; Mathieu et al., 2016; Bowman et al., 2015; Radford et al., 2015; Mescheder et al., 2017; Ha & Eck, 2017; Dinh et al., 2016; Larsen et al., 2015; Van den Oord et al., 2016; Lu et al., 2018; He et al., 2019). I.e., as we move from one latent vector to another, there are interpretable attributes of the outputs that gradually appear or disappear. This leads to the following verification question: given encodings of two outputs with a number of *shared* attributes, what fraction of the line segment between the encodings generates outputs sharing those same attributes? To anwer this question, we can verify a generator using a trusted attribute detector, or we verify an attribute detector based on a trusted generator. For both tasks, we have to analyze the outputs of neural networks restricted to line segments.

**EXACTLINE** Sotoudeh & Thakur (2019) computes succinct representations of a piecewise-linear neural networks restricted to such line segments $\overline{AB} \subset \mathbb{R}^m$. It is visualized in the top row of Figure 1, where a line segment $\overline{e_1 e_2} \subset \mathbb{R}^m$ between encodings produced by an encoder neural network $n_E$ is passed through a decoder $n_D$ and an attribute detector $n_A$. In more detail, the primitive $\mathcal{P}(N|_{\overline{AB}})$ computes a polygonal chain $(P_1, \ldots, P_k)$ in $\mathbb{R}^m$ representing the line segment $\overline{AB}$, such that the neural network $N$ is affine on the segment $\overline{P_i P_{i+1}}$ for all $0 \le i < k$. As a consequence, the polygonal chain $(N(P_1), \ldots, N(P_k))$ represents the image of $\overline{AB}$ under $N$.

To compute this, one can incrementally compute normalized distances on the input segment $\overline{AB}$ for the output of each layer $i$ of the network, $N_i$. Specifically, we find $0 = t_{i,1} \leq \cdots \leq t_{i,k_i} = 1$ such that $N_i$ is affine on $\overline{(A + t_{i,j}(A - B))(A + t_{i,j+1}(A - B))}$, and keep track of the nodes $N_i(A + t_{i,j}(A - B))$. In the case of affine operations such as matrix multiplication or convolution, one can simply apply that operation to each node and leave the distances unchanged. The case of ReLU more segments may be introduced. To compute ReLU, one can apply it per-dimension, $d$, and check for each $0 \leq j < k_i$ whether $N_i(A + t_{i,j}(A - B))_d < 0 < N_i(A + t_{i,j+1}(A - B))_d$. In this case, a new distance is computed: $t' = t_{i,j} + \frac{|N_i(A+t_{i,j}(A-B))_d|(t_{i,j+1}-t_j)}{|N_i(A+t_{i,j+1}(A-B))_d - N_i(A+t_{i,j}(A-B))_d|}$. This is done analogously in the case where the segment is decreasing in dimension $d$ instead of increasing.

We extend EXACTLINE to perform exact probabilistic inference by associating with each segment $P_j$ in the chain a probability distribution $\mu_j$ over that segment, and a probability $p_j$. In the case of the uniform distribution, as in Figure 1, every $\mu_j$ is also a uniform distribution, and $p_j = t_{j+1} - t_j$.

**APPROXLINE** Unfortunately, EXACTLINE sometimes scales poorly for tasks using generative models, because too many line segments are generated. We improve scaling by introducing a sound overapproximation, APPROXLINE. Instead of maintaining a single polygonal chain, APPROXLINE maintains a list of polygonal chains and a list of interval constraints[1], such that the neural network's activations are guaranteed to either lie on one of the polygonal chains or to satisfy one of the interval constraints. We introduce a novel *relaxation heuristic*, which chooses subsets of line segments in polygonal chains and replaces them with interval constraints that subsume them. Our relaxation heuristic attempts to combine line segments that are adjacent and short.

To perform probabilistic inference, each element also carries its probability. For a feed-forward network, each operation acts on the elements individually, without modifying the probability associated with it. This is shown in the bottom row of Figure 1. Here, the probabilities of the segments that get approximated are shown by the yellow regions in the top row. One can observe that they remain unchanged when converted to intervals (in blue) in the bottom row. One can also observe that probabilities associated with intervals do not change, even when the intervals change substantially. Python pseudocode for propogation of APPROXLINE is shown in Appendix A.

To understand the interaction between probabilistic inference and the relaxation heuristic, it is best to work through a contrived example. Suppose we have an EXACTLINE in two dimensions with the nodes $(1, 1), (2, 2), (3, 2), (5, 1), (7, 5)$ with weights $0.1, 0.1, 0.5, 0.3$ on its segments. The weights describe the probabilities of the output being on each segment respectively, where distributions on the segments themselves are uniform. Our relaxation heuristic might combine the line segments with nodes $(1, 1), (2, 2)$ and $(2, 2), (3, 2)$ and approximate them by a single interval constraint with lower bound $(1, 1)$ and upper bound $(3, 2)$. (I.e., the first component is between 1 and 3, and the second component is between 1 and 2.) In this case, we can understand the output of approximation as an APPROXLINE: An interval constraint with center $(2, 1.5)$, radius $(1, 0.5)$ and weight $0.1 + 0.1 = 0.2$, as well as a polygonal chain formed of the segment $\overline{(3, 2)(5, 1)}$ with weight $0.5$ and the segment $\overline{(5, 1)(7, 5)}$ with weight $0.3$. To compute a lower bound on the robustness probability, we would sum the probabilities of each element where it can be proven there is no violation. For example, assume that only the point $(1, 2)$ is disallowed by the specification. The inferred robustness probability lower bound would be $0.8$. To compute an upper bound on the robustness probability, we sum the probabilities of each element where it can be proven that at least one point is safe. Here, we obtain $1$.

## 3 RELATED WORK

Dvijotham et al. (2018a) verify probabilistic properties universally over sets of inputs by bounding the probability that a dual approach verifies the property. In contrast, our system verifies properties that are either universally quantified or probabilistic. However, the networks we verify are multiple orders of magnitude larger. While they only provide upper bounds on the probability that a specification has been violated, we provide extremely tight bounds on such probabilities from both sides. PROVEN (Weng et al., 2018) uses sampling to find high confidence bounds (confidence intervals) on the probability of misclassification. While PROVEN only provides high confidence bounds (99.99%), APPROXLINE provides bounds with 100% confidence. Nevertheless, our method is much faster and

---

[1]Lists of abstract domains are described later as Union and Powerset domains.

produces better results than a similar sampling-based technique for finding confidence intervals using Clopper & Pearson (1934) (used by smoothing methods). Another line of work is smoothing, which provides a defense with high confidence statistical robustness guarantees (Cohen et al., 2019; Lecuyer et al., 2018; Liu et al., 2018; Li et al., 2018; Cao & Gong, 2017). In contrast, APPROXLINE provides deterministic guarantees, and is not a defense.

# 4 REVIEW OF ABSTRACT INTERPRETATION

We briefly review important concepts, closely following their presentations given in previous work (Gehr et al., 2018) where applicable. In our work, we assume that we can decompose the neural network as a sequence of $l$ piecewise-linear layers: $N = L_l \circ \cdots \circ L_1$.

An abstract domain (Cousot & Cousot, 1977) is a set of symbolic representations of sets of program states. We write $\mathcal{A}_n$ to denote an abstract domain whose elements each represent an element of $\mathcal{P}(\mathbb{R}^n)$, in our case a set of vectors of $n$ neural network activations. The concretization function $\gamma_n : \mathcal{A}_n \to \mathcal{P}(\mathbb{R}^n)$ maps a symbolic representation $a \in \mathcal{A}_n$ to its concrete interpretation as a set $\mathbb{X} \in \mathcal{P}(\mathbb{R}^n)$ of neural network activation vectors.

The concrete transformer $T_f : \mathcal{P}(\mathbb{R}^m) \to \mathcal{P}(\mathbb{R}^n)$ of some function $f : \mathbb{R}^m \to \mathbb{R}^n$ maps subsets $\mathbb{X} \subseteq \mathbb{R}^m$ of the domain of $f$ to their image under $f$, i.e., $T_f(\mathbb{X}) = \{f(\boldsymbol{x}) \mid \boldsymbol{x} \in \mathbb{X}\}$. Using this notation, the $(\mathbb{X}, \mathbb{Y})$-robustness property of a neural network $N$ can be written as $T_N(\mathbb{X}) \subseteq \mathbb{Y}$.

An abstract transformer $T_f^{\#} : \mathcal{A}_m \to \mathcal{A}_n$ transforms symbolic representations to symbolic representations overapproximating the effect of the function $f : \mathbb{R}^m \to \mathbb{R}^n$, which means it is required to satisfy $T_f(\gamma_m(a)) \subseteq \gamma_n(T_f^{\#}(a))$ for all $a \in \mathcal{A}_m$.

Abstract transformers are compositional: For given functions $f : \mathbb{R}^t \to \mathbb{R}^n$ and $g : \mathbb{R}^m \to \mathbb{R}^t$ with abstract transformers $T_f^{\#} : \mathcal{A}_t \to \mathcal{A}_n$ and $T_g^{\#} : \mathcal{A}_m \to \mathcal{A}_t$, we can define an abstract transformer $T_{f \circ g}^{\#} : \mathcal{A}_m \to \mathcal{A}_n$ for their composition $f \circ g : \mathbb{R}^m \to \mathbb{R}^n$, namely $T_{f \circ g}^{\#} = T_f^{\#} \circ T_g^{\#}$. We will follow this recipe for the neural network $N$, abstracting it as $T_N^{\#} = T_{L_l}^{\#} \circ \cdots \circ T_{L_1}^{\#}$.

Abstract interpretation provides a sound, typically incomplete method to certify neural network robustness. Namely, to show that a neural network $N : \mathbb{R}^m \to \mathbb{R}^n$ is $(\mathbb{X}, \mathbb{Y})$-robust, it suffices to show that $\gamma_n(T_N^{\#}(a)) \subseteq \mathbb{Y}$, for some abstract element $a \in \mathcal{A}_m$ with $\mathbb{X} \subseteq \gamma_m(a)$.

**Box domain** An element of the box domain $\mathcal{B}_n$ is a pair of vectors $b = (\boldsymbol{c}, \boldsymbol{d})$ where $\boldsymbol{c}, \boldsymbol{d} \in \mathbb{R}^n$. The concretization function is $\gamma_n(\boldsymbol{c}, \boldsymbol{d}) = \{\boldsymbol{c} + \mathrm{diag}(\boldsymbol{d}) \cdot \beta \mid \beta \in [-1, 1]^n\}$. Abstract interpretation with the box domain $\mathcal{B}$ is equivalent to bounds propagation with standard interval arithmetic.

**Powerset domain** Given an abstract domain $\mathcal{A}$, elements of its powerset domain $\mathcal{P}(\mathcal{A})_n$ are (finite) sets of elements of $\mathcal{A}_n$. The concretization function is given by $\gamma_n(a) = \bigcup_{a' \in a} \gamma_n(a')$ (using the concretization function of the underlying domain $\mathcal{A}$). We can lift any abstract transformer for $\mathcal{A}$ to an abstract transformer for $\mathcal{P}(\mathcal{A})$ by applying the transformer to each of the elements.

**Union domain** Given abstract domains $\mathcal{A}$ and $\mathcal{A}'$, an element of their union domain is a tuple $(a, a')$ with $a \in \mathcal{A}_n$ and $a' \in \mathcal{A}'_n$. The concretization function is $\gamma_n(a, a') = \gamma_n(a) \cup \gamma_n(a')$. We can apply abstract transformers of the same function for $\mathcal{A}$ and $\mathcal{A}'$ to the tuple elements independently.

## 4.1 PROBABILISTIC ABSTRACT INTERPRETATION

We denote as $\mathbb{D}_n$ the set of probability measures over $\mathbb{R}^n$. Probabilistic abstract interpretation is an instantiation of abstract interpretation where deterministic points from $\mathbb{R}^n$ are replaced by measures from $\mathbb{D}_n$. I.e., a probabilistic abstract domain (Cousot & Monerau, 2012) is a set of symbolic representations of sets of measures over program states. We again use subscript notation to determine the number of activations: a probabilistic abstract domain $\mathcal{A}_n$ has elements that each represent an element of $\mathcal{P}(\mathbb{D}_n)$. The probabilistic concretization function $\gamma_n : \mathcal{A}_n \to \mathcal{P}(\mathbb{D}_n)$ maps each abstract element to the set of measures it represents.

For a measurable function $f\colon \mathbb{R}^m \to \mathbb{R}^n$, the corresponding probabilistic concrete transformer $T_f\colon \mathcal{P}(\mathbb{D}_m) \to \mathcal{P}(\mathbb{D}_n)$ maps a set of measures $\mathbb{M} \subseteq \mathbb{D}_m$ to a set of measures $\mathbb{M}' \subseteq \mathbb{D}_n$, given by

$$\mathbb{M}' = \left\{ \mathbb{Y} \mapsto \Pr_{\boldsymbol{x} \sim \mu}[f(\boldsymbol{x}) \in \mathbb{Y}] \,\middle|\, \mu \in \mathbb{M} \right\},$$

where $\mathbb{Y}$ ranges over measurable subsets of $\mathbb{R}^n$.

A probabilistic abstract transformer $T_f^\#\colon \mathcal{A}_m \to \mathcal{A}_n$ abstracts the probabilistic concrete transformer in the standard way: it satisfies $\forall a \in \mathcal{A}_m . \, T_f(\gamma_m(a)) \subseteq \gamma_n(T_f^\#(a))$, as in the deterministic setting.

Probabilistic abstract interpretation provides a sound method to compute bounds on robustness probabilities. Namely, to show that $\Pr_{\boldsymbol{x} \sim \mu}[N(\boldsymbol{x}) \in \mathbb{Y}] \in [l, u]$, it suffices to show that $\nu(\mathbb{Y}) \in [l, u]$ for each $\nu \in \gamma_n(T_N^\#(a))$ for some $a$ with $\mu \in \gamma_m(a)$.

**Domain lifting** Any deterministic abstract domain can be directly interpreted as a probabilistic abstract domain, where the concretization of an element is given as the set of probability measures whose support is a subset of the deterministic concretization. The original deterministic abstract transformers can still be used.

**Convex combinations** Given two probabilistic abstract domains $\mathcal{A}$ and $\mathcal{A}'$, we can form their convex combination domain, whose elements are tuples $(a, a', p)$ with $a \in \mathcal{A}_n$, $a' \in \mathcal{A}'_n$ and $p \in [0, 1]$. The concretization function is given by $\gamma_n(a, a', p) = \{(1 - p) \cdot \mu + p \cdot \mu' \mid \mu \in \gamma_n(a), \mu' \in \gamma_n(a')\}$. We can apply abstract transformers of the same function for $\mathcal{A}$ and $\mathcal{A}'$ to the respective elements of the tuple independently, leaving $p$ intact.

Similarly, given a single probabilistic abstract domain $\mathcal{A}$, elements of its convex combination domain are tuples $(a, \lambda)$ where $a \in \mathcal{A}_n^k$, $\lambda \in [0, 1]^k$ and $\sum_{i=1}^k \lambda_i = 1$ for some $k$. The concretization of an element is given by $\gamma_n(a, \lambda) = \{\sum_{i=1}^k \lambda_i \cdot \mu_i \mid \mu_i \in \gamma_n(a_i), i \in \{1, \ldots, k\}\}$. We can apply abstract transformers for $\mathcal{A}$ independently to each entry of $a$, leaving $\lambda$ intact.

## 5 APPROXLINE

Here we define APPROXLINE, its non-convex relaxations, and its usage for probabilistic inference.

### 5.1 DEFINITION AS AN ABSTRACT DOMAIN

First, note that we can use EXACTLINE to create an abstract domain $\mathcal{E}$. The elements of $\mathcal{E}_n$ are polygonal chains $(P_1, \ldots, P_k)$ in $\mathbb{R}^n$ for some $k$. The concretization function $\gamma_n$ maps a polygonal chain $(P_1, \ldots, P_k)$ in $\mathbb{R}^n$ to the set of points in $\mathbb{R}^n$ that lie on it. For a piecewise-linear function $f\colon \mathbb{R}^m \to \mathbb{R}^n$, its abstract transformer $T_f^\#\colon \mathcal{E}_m \to \mathcal{E}_n$ maps a polygonal chain $(P_1, \ldots, P_k)$ in $\mathbb{R}^m$ to a new polygonal chain in $\mathbb{R}^n$ by concatenating the results of the EXACTLINE primitive on consecutive line segments $\overline{P_i P_{i+1}}$, eliminating adjacent duplicate points and applying the function $f$ to all points. The resulting abstract transformers are exact, i.e., they satisfy the subset relation in $\forall a \in \mathcal{A}_m . \, T_f(\gamma_m(a)) \subseteq \gamma_n(T_f^\#(a))$ with equality.

Our abstract domain is the union of the powersets of the EXACTLINE and box domains. Therefore, an abstract element is a tuple of a set of polygonal paths and a set of boxes, whose interpretation is that the activations of the neural network in a given layer are on one of the polygonal paths or within one of the boxes. For $x_1, x_2 \in \mathbb{R}^n$, we write $S(x_1, x_2) = (\{(x_1, x_2)\}, \{\})$ to denote the abstract element that represents a single line segment connecting $x_1$ and $x_2$. Like EXACTLINE, we focus on the case where the abstract element describing the input activations captures such a line segment.

Note that if we use the standard lifting of abstract transformers $T_{L_i}^\#$ for the EXACTLINE and box domains into our union of powersets domain, propagating a segment $S(x_1, x_2)$ through the neural network $N = L_l \circ \cdots \circ L_1$ using the abstract transformer $T_N^\# = T_{L_l}^\# \circ \cdots \circ T_{L_1}^\#$ is equivalent to using only the EXACTLINE domain: As the standard lifting applies the abstract transformers to all elements of both sets independently, we will simply obtain an abstract element $(\{(P_1, \ldots, P_k)\}, \{\})$, where $(P_1, \ldots, P_k)$ is a polygonal path exactly describing the image of $\overline{x_1 x_2}$ under $N$.

**Relaxation**    Therefore, our abstract transformers may, before applying a lifted abstract transformer, apply relaxation operators that turn an abstract element $a$ into another abstract element $a'$ such that $\gamma_n(a) \subseteq \gamma_n(a')$. We use two kinds of relaxation operators: *bounding box* operators remove a single line segment, splitting the polygonal chain into at most two new polygonal chains (at most one on each side of the removed line segment). The removed line segment is then replaced by its bounding box. *Merge* operators replace multiple boxes by their common bounding box.

Carefully applying the relaxation operators, we can explore a rich tradeoff between the EXACTLINE domain and the box domain. Our analysis generalizes both: if we never apply any relaxation operators, the analysis reduces to EXACTLINE, and will be exact but potentially slow. If we relax the initial line segment into its bounding box, the analysis reduces to box and be will be imprecise but fast.

**Relaxation heuristic**    For our evaluation, we use the following relaxation heuristic, applied before each convolutional layer of the neural network. The heuristic is parameterized by a relaxation percentage $p \in [0, 1]$ and a clustering parameter $k \in \mathbb{N}$. Each chain with $t > 1000$ nodes is traversed from one end to the other, and each line segment is turned into its bounding box, until the chain ends, the total number of nodes visited exceeds $t/k$ or we find a line segment whose length is strictly above the $p$-th percentile, computed over all segment lengths in the chain prior to applying the heuristic. All bounding boxes generated in one such step (from adjacent line segments) are then merged, the next segment (if any) is skipped, and the traversal is restarted on the remaining segments of the chain. This way, each polygonal chain is split into some new polygonal chains and a number of new boxes.

## 5.2    PROBABILISTIC INFERENCE

The EXACTLINE domain can be extended such that it captures a single probability distribution on a polygonal chain. For each line segment $(P_i, P_{i+1})$ on the polygonal chain $(P_1, \ldots, P_k)$ in $\mathbb{R}^n$, we additionally store a symbolic representation of a measure $\mu_i$ on $[0, 1]$, such that $\sum_{i=1}^{k-1} \mu_i([0, 1]) = 1$. The abstract element $a = (P_1, \mu_1, P_2, \ldots, P_{k-1}, \mu_{k-1}, P_k)$ then represents the probability measure

$$\nu(\mathbb{X}) = \sum_{i=1}^{k-1} \mu_i \left( \left\{ \frac{\|\boldsymbol{x} - P_i\|_2}{\|P_{i+1} - P_i\|_2} \,\middle|\, \boldsymbol{x} \in \mathbb{X} \cap \overline{P_i P_{i+1}} \right\} \right),$$

where $\mathbb{X}$ ranges over measurable subsets of $\mathbb{R}^n$. I.e., we have $\gamma_n(a) = \{\nu\}$. Whenever an abstract transformer splits a line segment, it additionally splits the corresponding measure, appropriately applying affine transformations, such that the new measures each range over $[0, 1]$ again. Note that if measures are uniform, it suffices to store $\mu_i([0, 1])$ as the symbolic representation of $\mu_i$.

Our probabilistic abstract domain is the convex combination of the convex combination domains of this probabilistic EXACTLINE domain and the standard lifting of the box domain as a probabilistic abstract domain. In practice, it is convenient to store an abstract element $a$ with $p$ probabilistic polygonal chains and $q$ probabilistic boxes as

$$a = (((P_1^{(1)}, \mu_1^{(1)}, \ldots, \mu_{k_1-1}^{(1)}, P_{k_1}^{(1)}), \ldots, (P_1^{(p)}, \mu_1^{(p)}, \ldots, \mu_{k_p-1}^{(p)}, P_{k_p}^{(p)})), ((b^{(1)}, \ldots, b^{(q)}), \lambda)),$$

such that $\sum_{i=1}^{l} \sum_{j=1}^{k_i-1} \mu_j^{(i)}([0, 1]) + \sum_{i=1}^{q} \lambda_i = 1$. Its concretization is then given as

$$\gamma_n(a) = \left\{ \sum_{i=1}^{p} w_i \nu_i + \sum_{j=1}^{q} \lambda_i \nu_j' \,\middle|\, \nu_i \in \gamma_n \left( P_1^{(i)}, \mu_1^{(i)}/w_i, \ldots, \mu_{k_i-1}^{(i)}/w_i, P_{k_i}^{(i)} \right), \nu_j' \in \gamma_n(b^{(j)}) \right\},$$

where $w_i = \sum_{j=1}^{k_i-1} \mu_j^{(i)}([0, 1])$. Our input always captures a uniform distribution on a line segment.

**Relaxation and heuristic**    Our deterministic relaxations can be extended to work in the probabilistic setting. When we replace a line segment by its bounding box, we use the total weight in its measure as the new entry in the weight vector $\lambda$ corresponding to the box. When we merge multiple boxes, their weights are added to give the weight for the the resulting box. We then use the same relaxation heuristic as we described previously also in the probabilistic setting.

**Computing bounds**    Given a probabilistic abstract element $a$ as above, describing the output distribution of the neural network, we want to compute optimal bounds on the robustness probabilities $\mathbb{P} = \{\nu(\mathbb{Y}) \mid \nu \in \gamma_n(a)\}$. The part of the distribution tracked by the probabilistic

EXACTLINE domain has all its probability mass in perfectly determined locations, while the probability mass in each box can be located anywhere inside it. We can compute bounds $(l, u) = (\min \mathbb{P}, \max \mathbb{P}) = \left( e + \sum_{j \in \mathbb{L}} \lambda_j, e + \sum_{j \in \mathbb{U}} \lambda_j \right)$, where $e = \sum_{i=1}^{p} w_i \nu_i(\mathbb{Y})$, with $\{\nu_i\} = \gamma_n \left( P_1^{(i)}, \mu_1^{(i)}/w_i, \ldots, \mu_{k_i-1}^{(i)}/w_i, P_{k_i}^{(i)} \right)$, $\mathbb{L} = \{j \in \{1, \ldots, q\} \mid \gamma_n(b^{(j)}) \subseteq \mathbb{Y}\}$ and $\mathbb{U} = \{j \in \{1, \ldots, q\} \mid \gamma_n(b^{(j)}) \cap \mathbb{Y} \neq \emptyset\}$. Here, we used the *deterministic* box concretization $\gamma_n$.

## 6    EVALUATION

We write $\text{APPROXLINE}_k^p$ to denote our analysis (deterministic and probabilistic versions) where the relaxation heuristic uses relaxation percentage $p$ and clustering parameter $k$. We implement APPROXLINE as in the DiffAI framework, taking advantage of the GPU parallelization provided by PyTorch (Paszke et al., 2017). Additionally, we use our implementation of APPROXLINE to compute exact results without approximation. To get exact results, it suffices to set the relaxation percentage $p$ to 0, in which case the clustering parameter $k$ can be ignored. Verification using $\text{APPROXLINE}_k^0$ is equivalent to EXACTLINE up to floating point error. To distinguish our GPU implementation from the original CPU implementation, we call our method EXACT instead of EXACTLINE. EXACT is additionally capable of doing exact probabilistic inference. We run on a machine with a GeForce GTX 1080 with 12 GB of GPU memory, and four processors with a total of 64 GB of RAM.

### 6.1    GENERATIVE SPECIFICATIONS

For generative specifications, we use decoders from autoencoders with either $32$ or $64$ latent dimensions trained in two different ways: VAE and CycleAE, described below. We train them to reconstruct CelebA with image sizes $64 \times 64$. We always use Adam Kingma & Ba (2014) with a learning rate of $0.0001$ and a batch size of $100$. The specific network architectures are described in Appendix B. Our decoder always has $74128$ neurons and the attribute detector has $24676$ neurons.

**VAE$_l$** is a variational autoencoder (Kingma & Welling, 2013) with $l$ latent dimensions.

**CycleAE$_l$** is a repurposed CycleGAN (Zhu et al., 2017) with $l$ latent dimensions. While these were originally designed for unsupervised style transfer between two data distributions, $P$ and $Q$, we use it to build an autoencoder such that the generator behaves like a GAN and the encodings are distributed evenly among the latent space. Specifically, we use a normal distribution in $l$ dimensions for the embedding/latent space $P$ with a small feed forward network $D_P$ as the latent space discriminator. The distribution $Q$ is the image distribution, and for its discriminator $D_Q$ we use the BEGAN method (Berthelot et al., 2017), which determines an example's realism based on an autoencoder (also with $l$ latent dimensions), which is trained to reproduce the ground-truth distribution $Q$ and adaptively to fail to reproduce the GAN generator's distribution.

**Attribute Detector** is trained to recognize the $40$ attributes provided by CelebA. Specifically, the attribute detector has a linear output. We consider the attribute $i$ to be detected as present in the input image if and only if the $i$-th component of the output of the attribute detector is strictly greater than $0.5$. The attribute detector is trained using Adam, minimizing the L1 loss between either $1$ and the attribute (if it is present) or $0$ and the attribute (if it is absent). We train it for $300$ epochs.

### 6.2    SPEED AND PRECISION RESULTS

Given a generative model capable of producing interpolations between inputs which remain on the data manifold, there are many different verification goals one might pursue: E.g., check whether the generative model is correct with respect to a trusted classifier or whether a classifier is robust to interpretable interpolations between data points generated from a trusted generative model. Even trusting neither the generator nor the classifier, we might want to verify that they are consistent.

We address all of these goals by efficiently computing the *attribute consistency* of a generative model with respect to an attribute detector: For a point picked uniformly at random between the encodings $e_1$ and $e_2$ of two ground truth inputs with matching attributes, we would like to determine the probability that its decoding will have the same attribute $i$. We define the attribute consistency as

$$\mathcal{C}_{i, n_A, n_D}(e_1, e_2, t) = \Pr_{\alpha \sim U(0,1)} [n_A(n_D((1 - \alpha)e_1 + \alpha e_2))_i > 0.5 \iff t],$$

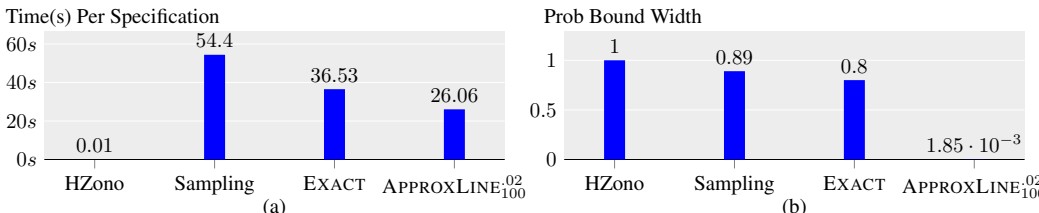

Figure 2: A comparison of different methods that compute probabilistic bounds for $\hat{\mathcal{C}}$.

where $t$ is the ground truth for attribute $i$. We will frequently omit the attribute detector $n_A$ and the decoder $n_D$ from $\mathcal{C}$ if it is clear from context which networks are being evaluated.

In this section, we demonstrate that probabilistic APPROXLINE is precise and efficient enough to provide useful bounds on the attribute consistency for interesting generative models and specifications on a reasonable dataset. To this end we compare APPROXLINE to a variety of other methods which are also capable of providing probabilistic bounds. We do this for CycleAE$_{32}$ trained for 200 epochs.

Specifically, suppose $P$ is a set of unordered pairs $\{\boldsymbol{a}, \boldsymbol{b}\}$ from the data set with $a_{A,i} > 0.5 \iff b_{A,i} > 0.5$ for each of the $k$ attributes $i$, where $a_A$ are ground truth attribute labels of $\boldsymbol{a}$. Using each method, we find bounds on the true value of *average attribute consistency* as $\hat{\mathcal{C}}_P(n_A, n_D, n_E) = \text{mean}_{\boldsymbol{a},\boldsymbol{b} \in P, i} \mathcal{C}_{i,n_A,n_D}(n_E(\boldsymbol{a}), n_E(\boldsymbol{b}), a_{A,i} > 0.5)$ where $n_E$ is the encoding network. Each method finds a probabilistic bound, $[l, u]$, such that $l \leq \hat{\mathcal{C}} \leq u$. We call $u - l$ its width.

We compare probabilistic APPROXLINE against two other probabilistic abstract domains, EXACT (=APPROXLINE$_k^0$), and HZono (Mirman et al., 2018) lifted probabilistically. Furthermore, we also compare against sampling with binomial confidence intervals on $\mathcal{C}$ using the ClopperPearson interval.

For probabilistic sampling, we take samples and recalculate the ClopperPearson interval with a confidence of 99.99% until the interval width is below 0.002 (chosen to be the same as our best result with APPROXLINE). To avoid an incorrect calculation, we discard this interval and prior samples, and resample using the estimated number of samples. Importantly, the probabilistic bound returned by the abstract domains is guaranteed to be correct 100% of the time, while for sampling it is only guaranteed to be correct 99.99% of the time.

For all methods, we set a timeout of 60s, and report the largest possible probabilistic bound if a timeout or out-of-memory error occurs. For APPROXLINE, if an out-of-memory error occurs, we refine the hyperparameters using schedule A in Appendix C and restart (without resetting the timeout clock). Figure 2 shows the results of running these on $|P| = 100$ pairs of matching celebrities with matching attribute labels, chosen uniformly at random from CelebA (each method uses the same $P$).

The graph shows that while HZono is the fastest domain, it is unable to prove any specifications. Sampling and EXACT do not appear to be significantly slower than APPROXLINE, but it can be observed that the average width of the probabilistic bounds they produce is large. This is because Sampling frequently times out, and EXACT frequently exhausts GPU memory. On the other hand, APPROXLINE provides an average probabilistic bound width of less than 0.002 in under 30s with perfect confidence (compared with the lower confidence provided by sampling).

## 6.3 USE CASES FOR APPROXLINE

Here, we demonstrate how to use our domain to check the attribute consistency of a model against an attribute detector. We do this for two possible generative specifications: (i) generating rotated heads using flipped images, and (ii) adding previously absent attributes to faces. For the results in this section, we use schedule B described in Appendix C.

**Comparing models with turning heads** It is known that VAEs are capable of generating images with intermediate poses of the subject from flipped images of the subject. An example of this transformation is shown in Figure 3b. Here, we show how one can use APPROXLINE to compare the effectiveness of different autoencoding models in performing this task. To do this, we trained all 4 architectures described above for 20 epochs. We then create a line specification over the encodings

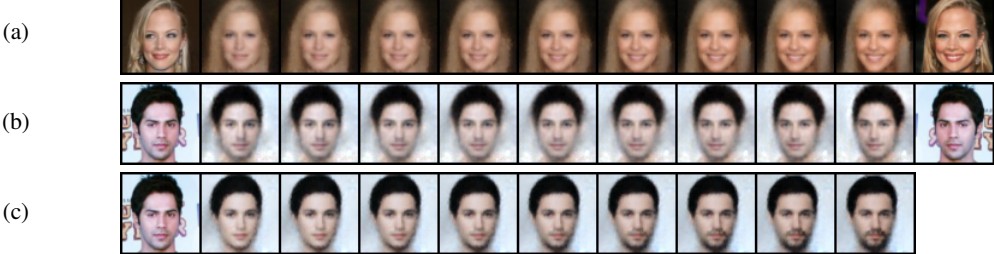

Figure 3: Examples of equally spaced interpolated images. The original images are at the 0th and 10th positions, and their immediate neighbors are their respective reconstructions. (a) is between the same person with the same attributes using CycleAE$_{32}$ trained for 200 epochs. (b) is between horizontally flipped images using CycleAE$_{64}$ trained for 20 epochs. (c) is between an image and the addition of the "mustache" feature vector using CycleAE$_{32}$ again.

Figure 4: Comparing the different models using probabilistic APPROXLINE$_{200}^{0.02}$ to provide lower bounds for $\frac{1}{k}\sum_{i=1}^{k}\mathcal{C}_i(n_E(\boldsymbol{a}), n_E(\text{Flipped}(\boldsymbol{a})), a_{A,i})$, where $\boldsymbol{a}$ and Flipped($\boldsymbol{a}$) are the images shown in Figure 3b. The width of the largest probabilistic bound was smaller than $3 \times 10^{-6}$, so only the lower bounds are shown. Less than 50 seconds were necessary to compute each bound, and the fastest computation was for CycleAE$_{64}$ at 30 seconds.

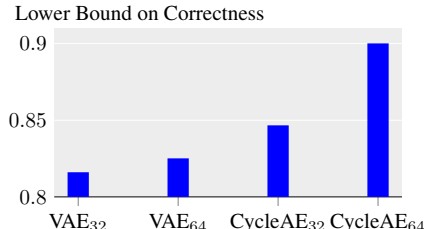

of the flipped images shown in Figure 3b. For a human face that is turned in one direction, ideally the different reconstructions will correspond to images of different orientations of the same face in 3D space. As none of the CelebA attributes correspond to pose, the attribute detector should recognize the same set of attributes for all interpolations. We used deterministic APPROXLINE$_{200}^{0.02}$ to demonstrate which attributes provably remain the correct for every possible interpolation (as visualized in Appendix E). While we are able to show in the worst case, 32 out of 40 attributes are entirely robust to flipping, some attributes are not robust across interpolation. Figure 4 demonstrates the results of using probabilistic APPROXLINE to find the average lower bound on the fraction of the input interpolation encodings which do result in the correct attribute appearing in the output image.

**Verifying attribute independence** Here, we demonstrate using APPROXLINE that attribute detection for one feature is invariant to a transformation in an independent feature. Specifically, we verify for a single image the effect of adding a mustache. This transformation is shown in Figure 3c. To do this, we find the attribute vector $m$ for "mustache" ($i = 22$ in CelebA) using the 80k training-set images in the manner described by Larsen et al. (2015), and compute probabilistic bounds for $\mathcal{C}_j(n_E(\boldsymbol{o}), n_E(\boldsymbol{o}) + 2m, o_{A,j})$ for $j \neq 22$ and the image $\boldsymbol{o}$. Using APPROXLINE we are able to prove that 30 out of the 40 attributes are entirely robust through the addition of a mustache. Among the attributes which can be proven to be robust are $i = 4$ for "bald" and $i = 39$ for "young". We are able to find that the attribute $i = 24$ for "NoBeard" is not entirely robust to the addition of the mustache vector. We find a lower bound on the robustness probability for that attribute of $0.83522$ and an upper bound of $0.83528$.

## 7    CONCLUSION

In this paper we presented a highly scalable non-convex relaxation to verify neural network properties where inputs are restricted to a line segment. Our results show that our method is faster and more precise than previous methods for the same networks, including sampling. This speed and precision permitted us to verify properties based on interesting visual transformations induced by generative networks for the first time, including probabilistic properties.

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

## A  APPROXLINE PROPOGATION PSEUDOCODE

**Algorithm 1:** Pseudocode for the APPROXLINE class. Here we only demonstrate propogation and not final computation of probabilities.

```
class ApproxLine ():
  def __init__(self, elements, probs):
    self.elements = elements
    self.probs = probs

  def multiply_add_relu_merge(self, matrix, bias):
    return ApproxLine([e.multiply_add_relu_merge() for e in self.elements()]
                      , self.probs)
```

**Algorithm 2:** Pseudocode for Interval propogation.

```
class Interval ():
  def __init__(self, a, b, center_radius = False):
    if center_radius:
      self.center = a
      self.radius = b
    else:
      self.center = (b + a) / 2
      self.radius = (b − a).abs() / 2

  def multiply_add_relu_merge(self, matrix, bias):
    center = self.center.multiply(matrix) + bias
    radius = self.radius.multiply(matrix.abs())
    return Interval(center, radius, self.prob)

  def expand(self, node):
    max_point = self.center + self.radius
    min_point = self.center − self.radius
    return Interval(self, minimum(max_point, node), maximum(max_point, node))
```

**Algorithm 3:** Pseudocode for the ExactLine class. This is not how it is implemented in either our framework or by Sotoudeh & Thakur (2019) but is useful for pedagogical purposes.

```
class ExactLine():
  def __init__(self, nodes, dists):
    self.nodes = nodes
    self.dists = dists
    assert(dists[-1] = 1)

  def multiply_add_relu_merge(self, matrix, bias):

    new_nodes = [ node * matrix + bias for node in self.nodes ]
    new_dists = self.dists

    # compute ReLUs inneficiently per dimension (pedagogically).
    for d in range(new_node[0].dimensions):
      relu_nodes = []
      relu_dists = []
      for n in range(len(new_nodes) - 1):
        curr_node = new_nodes[n]
        next_node = new_nodes[n + 1]

        curr_dist = new_dists[n]
        next_dist = new_dists[n + 1]

        dist_to_curr = next_dist - curr_dist
        relu_nodes += [ curr_node.relu() ]
        if (curr_node[d] < 0 and next_node[d] > 0)
         or (curr_node[d] > 0 and next_node[d] < 0):
          new_dist_to_curr =
            dist_to_curr * min(curr_node[d], next_node[d]).abs()
            / abs(next_node[d] - curr_node[d])
          relu_nodes +=
            [(next_node - curr_node)*new_dist_to_curr+curr_node]
          relu_dists += [curr_dist + new_dist_to_curr]
      relu_nodes += [next_node.relu()]
      relu_dists += [next_dist]

      new_nodes = relu_nodes
      new_dists = relu_dists

    # now cluster and merge
    return ExactLine([e.multiply_add_relu() for e in self.elements()]).merge()
```

**Algorithm 4:** Pedagogical pseudocode for merge (as part of the ExactLine class) based on a relaxation heuristic described in more detail in Section 5.1. This is not the most efficient possible implentation as is done for our framework. It is important to ensure that every edge becomes part of either an ExactLine or an Interval, and not counted twice.

```
def merge(self):
  cluster_map = heuristic(self)
  unclustered_segments = []
  unclustered_probs = []
  for i in range(len(new_nodes) - 1):
   if i in cluster_map.indexes_to_clusters:
     cluster =
       cluster_map.clusters[cluster_map.indexes_to_clusters[i]]
     cluster.interval.expand(self.nodes[i])
     cluster.interval.expand(self.nodes[i + 1])
     cluster_map.probs[cluster_map.indexes_to_clusters[i]] +=
       self.dists[i + 1] - self.dists[i]
    else:
     unclustered_segments +=
       [ExactLine(self.nodes[i], self.nodes[i+1])]
     unclustered_probs +=
       [self.dists[i+1] - self.nodes[i]]
  return ApproxLine(cluster_map.clusters + unclustered_segments
        ,cluster_map.probs + unclustered_probs)
```

## B  NETWORK ARCHITECTURES

For both models, we use the same encoders and decoders (even in the autoencoder descriminator from BEGAN), and always use the same attribute detectors. Here we use $\text{Conv}_s C \times W \times H$ to denote a convolution which produces $C$ channels, with a kernel width of $W$ pixels and height of $H$, with a stride of $s$ and padding of 1. FC $n$ is a fully connected layer which outputs $n$ neurons. $\text{ConvT}_{s,p} C \times W \times H$ is a transposed convolutional layer (Dumoulin & Visin, 2016) with a kernel width and height of $W$ and $H$ respectively and a stride of $s$ and padding of 1 and out-padding of $p$, which produces $C$ output channels.

- *Latent Descriminator* is a fully connected feed forward network with 5 hidden layers each of 100 dimensions.

- *Encoder* is a standard convolutional neural network:
  $x \rightarrow \text{Conv}_1 32 \times 3 \times 3 \rightarrow \text{ReLU} \rightarrow \text{Conv}_2 32 \times 4 \times 4 \rightarrow \text{ReLU} \rightarrow \text{Conv}_1 64 \times 3 \times 3$
  $\rightarrow \text{ReLU} \rightarrow \text{Conv}_2 64 \times 4 \times 4 \rightarrow \text{ReLU} \rightarrow \text{FC } 512 \rightarrow \text{ReLU} \rightarrow \text{FC } 512 \rightarrow l.$

- *Decoder* is a transposed convolutional network which has 74128 neurons:
  $l \rightarrow \text{FC } 400 \rightarrow \text{ReLU} \rightarrow \text{FC } 2048 \rightarrow \text{ReLU}$
  $\rightarrow \text{ConvT}_{2,1} 16 \times 3 \times 3 \rightarrow \text{ReLU} \rightarrow \text{ConvT}_{1,0} 3 \times 3 \times 3 \rightarrow x$

- *Attribute Detector* has 24676 neurons:
  $x \rightarrow \text{Conv}_2 16 \times 4 \times 4 \rightarrow \text{ReLU} \rightarrow \text{Conv}_2 32 \times 4 \times 4 \rightarrow \text{ReLU} \rightarrow \text{FC } 100 \rightarrow 40.$

## C  APPROXLINE REFINEMENT SCHEDULE

While many refinement schemes start with an imprecise approximation and progressively tighten it, we observe that being only occasionally memory limited and rarely time limited, it conserves more time to start with the most precise approximation we have determined usually works, and progressively try less precise approximations as we determine that more precise ones can not fit into GPU memory. Thus, we start searching for a probabilistic robustness bound with $\text{APPROXLINE}_N^p$ and if we run out of memory, try $\text{APPROXLINE}_{\max(0.95N,5)}^{\min(1.5p,1)}$ for schedule A, and $\text{APPROXLINE}_{\max(0.95N,5)}^{\min(3p,1)}$ for schedule B. This procedure is repeated until a solution is found, or time has run out.

## D  EFFECT OF APPROXIMATION PARAMETERS ON SPEED AND PRECISION

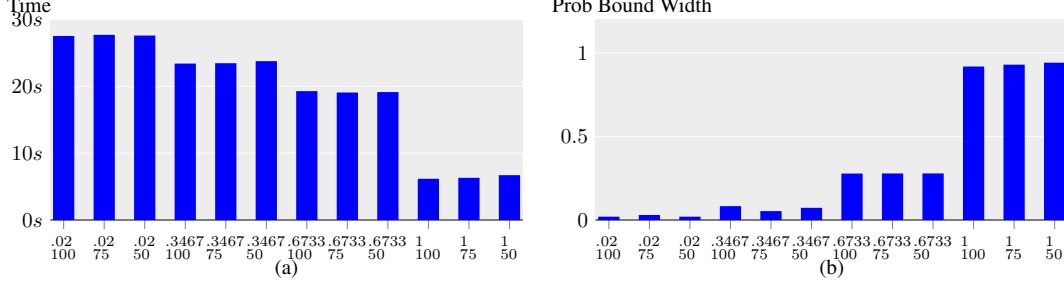

Figure 5: A comparison of speed (a) and Probabilistic Bound Widths (b) of APPROXLINE for different approximation hyperparameters, on $\text{CycleAE}_6 4$ trained for 200 epochs.

Here we demonstrate how modifying the approximation parameters, $p$ and $N$ of $\text{APPROXLINE}_N^p$ effect its speed and precision. Figure 5 shows the result of varying these on x-axis. The bottom number, $N$ is the number of clusters that will be ideally made, and the top number $p$ is the percentage of nodes which are permitted to be clustered.

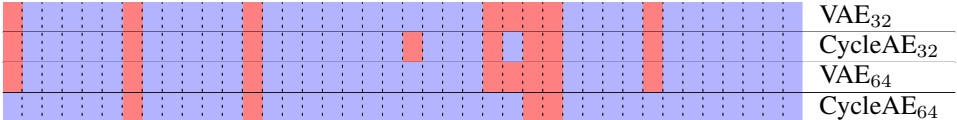

Figure 6: Blue means that the interpolative specification visualized in Figure 3b has been deterministically and entirely verified for the attribute (horizontal) using APPROXLINE$_{200}^{0.02}$. Red means that the attribute can not be verified. In all cases, this is because the specification was not robust for the attribute. One can observe that the most successful autoencoder is CycleAE$_{64}$.

## E  COMPARING THE DETERMINISTIC BINARY ROBUSTNESS OF DIFFERENT MODELS

Figure 6 uses deterministic APPROXLINE$_{200}^{0.02}$ to demonstrate which attributes provably remain the same and are correct (in blue) for every possible interpolation.

