# OpenReview forum: "Verification of Generative-Model-Based Visual Transformations"
_ICLR.cc/2020/Conference — Reject_

### Official Review · AnonReviewer3 · 2019-10-18
**Official Blind Review #3**

**Rating:** 3

**Review:**

Summary

This work aims to provide warranties on the outputs of generative models by providing bounds on robustness (over adversarial attacks for instance, or other transformation in this case). The specific case of restricting the inputs to a line segment allows performing verification of robustness exactly (Exact-line approach, NeurIPS'19). The authors extend this work and apply it to verify robustness of some VAE and BEGAN like models.

Positive aspects:
- Rigorous work, I did not spot much typos.
- First proofs given for generative models.

Negative aspects:

My main concern about this work is that the presentation is not didactic enough.
- From the beginning, key concepts are not clearly defined, such as network certification, specification, and the "verification problem". In the definition of robustness, a reference to a "safe set of outputs", as in Gehr et al. would help the understanding.
- The introduction is too short and lacks context. Figure 1 is not referred in the text and is not understandable with notations that are not yet introduced.
- Then follows without transition two pages of background that are mostly definitions but without a proper motivation, these are difficult to process.
- The work, an extension of the Exact-line approach, only gives a 5 lines description which is not insufficient to understand the approach.

Perhaps my assessment is too negative because I am unfamiliar with the certification literature, but since the work present applications in generative modeling, I think it should be understandable by readers from this background as well.
Minor:
of of in caption of Fig 2
last line of page 5: J -> j
I found the equivalence sign in the last equation of page 6 confusing, is it really supposed to be an equivalence here?
Page 6 : attribute detector... described below -> in appendix

After seeing the authors changes in the manuscript the paper does look better, but similarly to Reviewer 2 I still judge it difficult to understand.

**Experience Assessment:**

I do not know much about this area.

**Review Assessment: Checking Correctness Of Derivations And Theory:**

I assessed the sensibility of the derivations and theory.

**Review Assessment: Checking Correctness Of Experiments:**

I assessed the sensibility of the experiments.

**Review Assessment: Thoroughness In Paper Reading:**

I read the paper thoroughly.

---

> ### Author Response · Authors · 2019-11-12
> **Response**
>
> We thank you for your review, and have fixed the typos that have been spotted.  While Figure 1 is referred to on the first page in the original version, we agree that this is insufficient and have added an overview which we hope explains it and our methods better.  As per your suggestion, we included these relevant definitions, and added pseudocode which will hopefully help explain our extensions to ExactLine.

---

### Official Review · AnonReviewer2 · 2019-10-23
**Official Blind Review #2**

**Rating:** 3

**Review:**

summary:
The paper proposes a method to efficiently verify that generative models are consistent with respect to some known (latent) attribute. The authors defines attribute consistency by 1) mapping pairs of input (x1, x2) with matching attribute to a latent space using an encoder n_E(x) and 2) measuring how correctly an auxiliary classifier will classify the known attribute using (decoded) linear interpolations between the two latent encodings. Importantly, the proposed method gives guaranteed bounds on this consistency score, as opposed to simply evaluating the classifier on a fixed set uniformly sampled points between x1 and x2. In experiments the authors use their method to test for attribute independence as well as consistency under left-right flipping of an image using two different autoencoder models (VAE and CycleAE) obtaining tighter bounds on the ‘attribute consistency’ score than competing methods.

Decision & supporting arguments:
Conceptually I found the paper very appealing, and it tackles an important problem in generative modelling. However I have some concerns with respect to the paper in its current state:
1) It is not clear to me why the attribute consistency score, a key component in the paper, is a good measure of consistency in generative models. Notably, I miss motivation for why linear interpolations between encoded inputs should necessarily keep the attribute stable.
2) Although I found the experiments interesting, I did not find the experimental section completely comprehensive. There is no discussion or experiments probing the dependency on the quality of the auxiliary classifier or the encoder/decoder model used.
3) I did not find the description of the proposed method to be reasonably self-contained. Especially section 3 which describes the proposed method is challenging to follow. The background material in section 2 reads very much like a set of definitions. Since ICLR has a quite broad audience, I think the paper should be written in a more pedagogical way, with for instance clarifying examples. An example of a sentence that is incredibly hard to parse is on page 4, describing domain lifting: “Any deterministic abstract domain can be directly interpreted as a probabilistic abstract domain, where the concretization of an element is given as the set of probability measures whose support is a subset of the set produced by the deterministic concretization.” I think making this paper more pedagogical requires major rewriting.

Due to the above reasons I currently score the paper as a ‘weak reject’.

Further detailed questions/comments:
Consistency Score
Q1: What is the motivation behind the definition of the consistency attribute score. Especially, why is an attribute considered consistent if it is stable to linear interpolations in the latent space?

Experiment Results:
Q2.1: Did you perform any experiments on how the L1 score of the auxiliary classifier affects the consistency score? I would also like to see some quantitative numbers on the auxiliary classifier.
Q2.2: Why is the L1 score used for training the classifier instead of bernoulli which seems more natural for binary attributes?
Q2.3: Similarly, I would like to see some numbers on the quality of the encoder/decodes. Simply inspecting the interpolations in figure 3) the reconstructions seem quite blurry, likely due to the relatively small models used. Is it prohibitively expensive to run the proposed method on bigger models (e.g. ResNet based encoder/decoders or Unet-style models)?
Q2.4: I believe it would be more informative to show the actual confidence intervals in figure 2b) instead of only the width of the confidence intervals?

Readability:
Q3: I found it quite challenging to understand how the proposed is implemented in practice - My suggestion is that the authors add a pseudo-code / algorithm to section 3 clarifying exactly how the bounds reported in the experimental section are calculated.



**Experience Assessment:**

I do not know much about this area.

**Review Assessment: Checking Correctness Of Derivations And Theory:**

I did not assess the derivations or theory.

**Review Assessment: Checking Correctness Of Experiments:**

I carefully checked the experiments.

**Review Assessment: Thoroughness In Paper Reading:**

I read the paper at least twice and used my best judgement in assessing the paper.

---

> ### Author Response · Authors · 2019-11-12
> **Response Part 2**
>
> Q2.0:  There is no discussion or experiments probing the dependency on the quality of the auxiliary classifier or the encoder/decoder model used.
>
> We have in fact included experiments comparing different autoencoder models using a single auxiliary classifier (Figure 4).  We can also certainly include experiments comparing different auxiliary classifiers.  However, we note that the goal of our experimental section is to highlight the performance of our certification method and didactically demonstrate the variety of properties it is capable of certifying, and not to make claims about the full performance of models. Our hope is that our system might be used as yet another tool to evaluate these systems, rather than as a definitive answer to the superiority of any system at the moment.
>
> Q2.1: Did you perform any experiments on how the L1 score of the auxiliary classifier affects the consistency score? I would also like to see some quantitative numbers on the auxiliary classifier.
>
> Do you mean F1 Score?  We note that our contribution is to the efficiency of a technique for certifying models and not a judgement on whether the models are actually quality models. However, we would be happy to run such tests and include scores for any (relu) models that you think would be of interest to the community.
>
> Q2.3: Similarly, I would like to see some numbers on the quality of the encoder/decodes. Simply inspecting the interpolations in figure 3) the reconstructions seem quite blurry, likely due to the relatively small models used. Is it prohibitively expensive to run the proposed method on bigger models (e.g. ResNet based encoder/decoders or Unet-style models)?
>
> It is true that our system is somewhat size-limited, but more relevantly to the models mentioned, our system is currently limited to feed-forward connections. While this is a practical limitation and not a theoretical limitation (DiffAI and ERAN can both handle recurrent connections), the non-convexity of the domain would mean that such connections would introduce multiplicatively many overapproximations. Specifically, n convex components (line segments or boxes) from a previous layer added to m convex components from a downstream layer would produce n*m boxes. While it is an interesting future research objective to optimize this case, it is out of the scope of this work.
>
> However, we would be happy to run our system on any feed-forward relu models, within an order of magnitude of the size we have shown, for any papers that you would like (especially if code is available).
>
> Q2.4: I believe it would be more informative to show the actual confidence intervals in figure 2b) instead of only the width of the confidence intervals?
>
> The purpose of Figure 2b is only to demonstrate the relative performance of the different certification techniques. The confidence interval width does not refer to an estimate on the correct output of these methods, but to the comparable property of the output of these methods themselves. We are happy to put the actual confidence intervals in the appendix, but for HZono, Sampling, and Exact, these would be almost entirely vacuous (as the confidence interval widths are so large), and for ApproxLine it would be indistinguishable from a line in the middle of the graph, which we worry would be a confusing visualization to have in the main body when comparing certification methods.
>
> Q3: I found it quite challenging to understand how the proposed is implemented in practice - My suggestion is that the authors add a pseudo-code / algorithm to section 3 clarifying exactly how the bounds reported in the experimental section are calculated.
>
> We have updated the paper to include pseudocode and have added an overview which explains our methods at a high level (we already updated the overview).  We will continue to lighten the presentation of ApproxLine to reduce the dependency on prior abstract interpretation knowledge.

---

> > ### Comment · AnonReviewer2 · 2019-11-15
> > **Response to rebuttal**
> >
> > I appreciate that the authors added the overview section, it benefits the paper. However, I stand by my opinion that the paper is still too hard to read for the target audience at ICLR.  Specifically section 3 and 4 are still challenging to follow for readers without extensive knowledge in program verification with sentences like:
> >
> > “Any deterministic abstract domain can be directly interpreted as a probabilistic abstract domain, where the concretization of an element is given as the set of probability measures whose support is a subset of the deterministic concretization.” (section 4)
> >
> > Or
> >
> > “Therefore, our abstract transformers may, before applying a lifted abstract transformer, apply relaxation operators that turn an abstract element a into another abstract element a’ such that ….” (section 5)
> >
> >
> > I believe the paper contains several interesting ideas, however the presentation of the contributions simply makes it too hard to read in its current state.

---

> ### Author Response · Authors · 2019-11-12
> **Response Part 1**
>
> Thank you for the thorough and clear review.  We will answer in two parts.
>
> Q1.1: It is not clear to me why the attribute consistency score, a key component in the paper, is a good measure of consistency in generative models. Notably, I miss motivation for why linear interpolations between encoded inputs should necessarily keep the attribute stable.
>
> There have been a wealth of papers that propose autoencoder/generative model systems that claim that linear interpolations produce "interpretable" results [1-11]. It is in fact possible that not all defined attributes for a dataset should be preserved under "interpretable" interpolations. For example, interpolating between a person with only a beard and the same person with only a mustache would likely fail to satisfy the disjunctive attribute "no beard or no mustache." However, for many attributes we do expect intuitively consistency along interpolations between examples with those attributes, such as "has blond hair." One can examine the named attributes provided with CelebA to decide whether they should remain consistent among interpolations (we believe they should).
>
> Interestingly, this brings up the important point that deciding whether an attribute should correspond to a direction in the encoded representation for a dataset is likely a subjective question. We do not claim to answer this. Rather, our system attempts to verify this for a given dataset and given property.
>
> Q1.2: Especially, why is an attribute considered consistent if it is stable to linear interpolations in the latent space?
>
> We clarify that we do not measure the consistency of an attribute in vacuum, but with respect to a particular autoencoder. An attribute which is consistent for one autoencoder might very well be inconsistent for another autoencoder, and this is not a judgement on the consistency of the attribute as much as it is a judgement on the consistency of the autoencoder.
>
> [1] Vincent Dumoulin, Ishmael Belghazi, Ben Poole, Olivier Mastropietro, Alex Lamb, Martin Arjovsky, and Aaron Courville. Adversarially learned inference. arXiv preprint arXiv:1606.00704, 2016.
> [2] Michael F Mathieu, Junbo Jake Zhao, Junbo Zhao, Aditya Ramesh, Pablo Sprechmann, and Yann LeCun. Disentangling factors of variation in deep representation using adversarial training. In Advances in Neural Information Processing Systems, pp. 5040–5048, 2016.
> [3] Samuel R Bowman, Luke Vilnis, Oriol Vinyals, Andrew M Dai, Rafal Jozefowicz, and Samy Bengio.
> Generating sentences from a continuous space. arXiv preprint arXiv:1511.06349, 2015.
> [4] Alec Radford, Luke Metz, and Soumith Chintala. Unsupervised representation learning with deep convolutional generative adversarial networks. arXiv preprint arXiv:1511.06434, 2015.
> [5] Lars Mescheder, Sebastian Nowozin, and Andreas Geiger. Adversarial variational bayes: Unifying variational autoencoders and generative adversarial networks. In Proceedings of the 34th International Conference on Machine Learning-Volume 70, pp. 2391–2400. JMLR. org, 2017.
> [6] David Ha and Douglas Eck. A neural representation of sketch drawings. arXiv preprint arXiv:1704.03477, 2017.
> [7] Laurent Dinh, Jascha Sohl-Dickstein, and Samy Bengio. Density estimation using real nvp. arXiv preprint arXiv:1605.08803, 2016.
> [8] Anders Boesen Lindbo Larsen, Søren Kaae Sønderby, Hugo Larochelle, and Ole Winther. Autoencoding beyond pixels using a learned similarity metric. arXiv preprint arXiv:1512.09300, 2015.
> [9] Aaron Van den Oord, Nal Kalchbrenner, Lasse Espeholt, Oriol Vinyals, Alex Graves, et al. Conditional image generation with pixelcnn decoders. In Advances in neural information processing systems, pp. 4790–4798, 2016.
> [10] Yongyi Lu, Yu-Wing Tai, and Chi-Keung Tang. Attribute-guided face generation using conditional cyclegan. In Proceedings of the European Conference on Computer Vision (ECCV), pp. 282–297, 2018.
> [11] Zhenliang He, Wangmeng Zuo, Meina Kan, Shiguang Shan, and Xilin Chen. Attgan: Facial attribute editing by only changing what you want. IEEE Transactions on Image Processing, 2019.

---

### Official Review · AnonReviewer1 · 2019-10-24
**Official Blind Review #1**

**Rating:** 6

**Review:**

This paper proposes APPROXLINE, which is a sound approximation to EXACTLINE and is able to compute tight deterministic bounds on probabilities efficiently when the input is restricted on a line. It is a nonconvex relaxation, therefore it is able to capture the nonconvexity of neural networks. APPROXLINE is applied to generative models to verify the consistency of image attributes through linear interpolations on the latent variables.

To me, the most significant part is that the proposed approach has the potential to become a reliable metric for evaluating whether a generator disentangles latent representations, as long as a reliable attribute classifier can be trained. I would suggest the authors to emphasize this part in their future versions.

However, the current version is quite difficult for me to understand, and I guess it is difficult for a broad range of audiences without background in program analysis. Somehow I think the same message can be conveyed better without abusing terms from abstract interpretation. I would also suggest the authors to reduce such abuse of notations. At least, pseudocode could be provided.

As a result, I cannot give a confident judgement about whether the contribution of this paper is significant given the existence of EXACTLINE. Still, I tend to accept this paper for its potential to become a good metric for generative models.

**Experience Assessment:**

I have published one or two papers in this area.

**Review Assessment: Checking Correctness Of Derivations And Theory:**

I did not assess the derivations or theory.

**Review Assessment: Checking Correctness Of Experiments:**

I assessed the sensibility of the experiments.

**Review Assessment: Thoroughness In Paper Reading:**

I made a quick assessment of this paper.

---

> ### Author Response · Authors · 2019-11-12
> **Significance**
>
> Thank you for your review. We will gladly incorporate your suggestions to reduce our reliance on terms and knowledge from abstract interpretation, and have included pseudocode so that our techniques can be understood in isolation.  As probabilistic abstract interpretation has proved a powerful framework for probabilistic bound inference in the program analysis community, we felt that it would be an important contribution on its own to describe a usage for the machine learning community.
>
> On the significance over ExactLine:  providing fast and sound overapproximations to exact methods is known to be non-trivial, and certainly publishable (see [1,2,3,4,5,6,7,8]). While abstracting exact domains to less precise domains (sets of boxes) is a well known technique, the decision of whether, where, when, and what to approximate has no definitive answer. In the case of restrictions to lines, we found that for larger networks, to be significantly more performant than simply sampling (as can be seen in Figure 2 and Section 4.2), over-approximation of ExactLine was absolutely necessary. The design of our core novel contribution, a heuristic for the parts of ExactLine to approximate, turned out to be both a delicate and a critical problem. We tested quite a few seemingly obvious heuristics, such as clustering ExactLine edges using k-means (both in the original dimensional space and projected to single dimensions in various ways) before determining the necessity of ensuring connectedness between the nodes in each approximation cluster (which turned out to be the best heuristic, and the one we present). We will update the paper to mention these other options.
>
> [1] Singh Gagandeep, Gehr Timon, Püschel Markus, Vechev Martin. Boosting Robustness Certification of Neural Networks.  ICLR 2019
> [2] Singh Gagandeep, Gehr Timon, Mirman Matthew, Püschel Markus, Vechev Martin.
> Fast and Effective Robustness Certification. NeurIPS 2018
> [3] Singh Gagandeep, Ganvir Rupanshu, Püschel Markus, Vechev Martin. Beyond the Single Neuron Convex Barrier for Neural Network Certification. NeurIPS 2019
> [4] Zhang, Huan, Tsui-Wei Weng, Pin-Yu Chen, Cho-Jui Hsieh, and Luca Daniel. Efficient neural network robustness certification with general activation functions. NeurIPS 2018.
> [5] Wang, Shiqi, Kexin Pei, Justin Whitehouse, Junfeng Yang, and Suman Jana. Efficient formal safety analysis of neural networks. In NeurIPS 2018.
> [6] Salman, Hadi, Greg Yang, Huan Zhang, Cho-Jui Hsieh, and Pengchuan Zhang. A convex relaxation barrier to tight robust verification of neural networks. NeurIPS 2019.
> [7] Wong, Eric and J. Z. Kolter. Provable Defenses against Adversarial Examples via the Convex Outer Adversarial Polytope. ICML 2018
> [8] Mirman Matthew, Gehr Timon, and Vechev Martin. Differentiable Abstract Interpretation for Provably Robust Neural Networks. ICML 2018

---

### Author Response · Authors · 2019-11-12
**General Response**

We thank all reviewers for their detailed responses.  A common thread among all reviewers is that our presentation was not pedagogical enough or self contained given the target audience.

By now various prior works published at ICLR [1], NeurIPS [3,4], and ICML [2] have heavily relied on abstract interpretation for verification and training and have achieved state of the art results.

To our knowledge, this is the first time probabilistic abstract interpretation has been introduced in this context, and we believe it can have similar impact as in classic verification. As this is the first time this method was introduced in this context, the presentation was not perfectly accessible, even though all definitions required to technically understand our work are present.

Because of this, we will work very hard on improving the presentation -- we have already updated the overview to provide for a more intuitive presentation. We will update the introduction and the paper to provide even more intuition and simplify our presentation by shifting some of the terms to appendix, explaining them less formally in the text, and reduce the overhead needed to understand key concepts.

[1] Singh Gagandeep, Gehr Timon, Püschel Markus, and Vechev Martin. Boosting Robustness Certification of Neural Networks.  ICLR 2019
[2] Mirman Matthew, Gehr Timon, and Vechev Martin. Differentiable Abstract Interpretation for Provably Robust Neural Networks. ICML 2018
[3] Singh Gagandeep, Gehr Timon, Mirman Matthew, Püschel Markus, and Vechev Martin.
Fast and Effective Robustness Certification. NeurIPS 2018
[4] Singh Gagandeep, Ganvir Rupanshu, Püschel Markus, and Vechev Martin. Beyond the Single Neuron Convex Barrier for Neural Network Certification. NeurIPS 2019

---

### Decision · Program_Chairs · 2019-12-19

**Decision:**

Reject

**Comment:**

The goal of verification of properties of generative models is very interesting and the contributions of this work seem to make some progress in this context. However, the current state of the paper (particularly, its presentation) makes it difficult to recommend its acceptance.